# Efficacy of various survey methods to detect an experimental population of spot-tailed earless lizards: A case study

Evan Drake Rangel[1]☯, Scott E. Henke (ID)[1]☯*, David B. Wester[1]☯, Gabriel Andrade-Ponce (ID)[2]☯, Ruby A. Ayala[3]☯, Cord B. Eversole[2]☯

1 Caesar Kleberg Wildlife Research Institute, Texas A&M University-Kingsville, Kingsville, Texas, United States of America, 2 Arthur Temple College of Forestry and Agriculture, Stephen F. Austin State University, Nacogdoches, Texas, United States of America, 3 Department of Biology and Chemistry, Texas A&M International University, Laredo, Texas, United States of America

☯ These authors contributed equally to this work.
* scott.henke@tamuk.edu

## Abstract

Plateau (*Holbrookia lacerata*) and Tamaulipan (*Holbrookia subcaudalis*) spot-tailed earless lizard (STEL) populations have experienced declines in population size and distribution. Both species are considered species of concern and Tamaulipan STEL are being considered for federal threatened status. Even with this heightened concern, little is known about these species. Therefore, our objectives were to determine the most effective and time-efficient methods to survey for STEL, and to determine if a lizard density threshold was required before STEL presence could be detected. We evaluated ten standard reptile search techniques (i.e., pitfall traps, funnel traps, two thermoregulation lures (i.e., rock mounds and cover boards), remote camera surveys, detection dog surveys, quadrant searches, systematic visual searches, environmental DNA sampling, and road cruising) to identify STEL relative abundance within a 1 ha enclosure. The 1 ha enclosure was divided into 100, 10 x 10 m quadrants and each reptile search technique was replicated five times and randomly assigned to a quadrant without replacement. STEL were randomly placed inside the enclosure at known densities of 5, 10, 20, 30, and 40 lizards per ha and their relative abundance was assessed by each method three times at each STEL density during August – September, 2021. STEL were allowed 6-day acclimation periods before increasing density. Because STEL were translocated to novel habitat, caution in interpretation should be noted. However, STEL were not detected using funnel traps, rock mounds, cover boards, remote cameras, and detector dogs at any density level. Pitfall traps, quadrant searches, and eDNA samples detected few STEL, but only at 40 STEL/ha density. Only systematic visual searches and road cruising yielded STEL detections at multiple densities; however, neither method could reliably predict STEL density. Because our detection rates were low (~7% at any STEL density), road cruising can be more time efficient to survey a larger area. Once locations with STEL have been

**Data availability statement:** All relevant data are within the paper.

**Funding:** Texas Comptroller of Public Accounts.

**Competing interests:** The authors have declared that no competing interests exist.

identified, then systematic visual searches between 1300–1500 hr can be conducted to determine the relative abundance of these elusive species.

## Introduction

Spot-tailed earless lizards (STEL) were recently separated into two distinct species, the Plateau STEL (*Holbrookia lacerata*) and the Tamaulipan STEL (*Holbrookia subcaudalis*) [1,2]. Spot-tailed earless lizards were initially considered a single species represented by two subspecies (*Holbrookia lacerata lacerata* and *H. l. subcaudalis*) [3] and occurred from coastal Texas near the Corpus Christi Bay, north to Austin, extended westward to Midland, Texas, and southward to include northeastern Mexico [3]. However, recent studies have determined that the Balcones Escarpment fault line separates the northern Plateau STEL (*H. lacerata*) from the southern Tamaulipan (*H. subcaudalis*) populations [2].

Both species have experienced sharp declines in their abundance and distribution [4], which is thought to have led to local extinctions across their historic ranges [1]. For example, Tamaulipan STEL historically were considered to occur throughout ~15.4 M ha of southern Texas (https://www.texascounties.net/statistics/region.htm), but today, only three isolated and scattered populations, constituting a total area of about 2000 ha, are known to occur [5]. Because the known current distribution of Tamaulipan STEL is estimated to be 0.0001% of its historic range, the United States Fish and Wildlife Service (USFWS) is considering threatened status within the Endangered Species Act (ESA) for the species [6]. By comparison, populations of the Plateau STEL also are scattered, mainly occur in central Texas, but the USFWS [7] recently decided that the Plateau STEL does not warrant listing as a federally threatened species, citing their current distributions, although reduced from historical records, appear stable and not likely to become endangered within the foreseeable future. Hypotheses for the decline of both species include pesticides, invasive fauna, and the invasion of exotic grasses [8]. In addition, agricultural practices and urbanization have been suggested as potential factors in the decline [9]; however, Axtell [3] deemed anthropogenic habitat modifications as advantageous to the species. Proponents against the ESA listing argued that too little was known about either species and their distribution to make such a determination [10].

Spot-tailed earless lizards are small, cryptic, elusive, and seemingly rare lizards that spend much time underground [11]. Recapture rates of marked STEL have been low regardless of year or survey method [4]. Axtell [12,13] reported boom and bust cycles in STEL populations. Duran [14] noted he found numerous STEL in 2015, but was unable to locate a single individual in the same locations the following year. Because STEL abundances and distributions have declined, and that both species have been difficult to consistently locate, an assessment by the Texas Comptroller Office was requested; however, it was unknown as to which assessment method was best to conduct a STEL survey.

Lizard survey techniques have included active methods, such as visual searches, road cruising, and use of artificial structures like cover boards and rock mounds to

entice reptiles for thermoregulation purposes, and passive methods, such as remote cameras, pitfall traps, and funnel traps [15]. Newer techniques, such as the use of detector dogs to locate reptiles by olfaction [16], and environmental DNA, where reptiles leave behind traces of the DNA in soils on which they transverse [17], are becoming more prominent in scientific literature. However, of these various survey techniques, a dearth of information exists concerning STEL. Hibbitts et al. [11] suggested that road cruising for STEL would transverse the greatest acreage in a shorter period of time. Garden et al. [18] stated that pitfall traps and visual searches were the most cost-effective methods to detect terrestrial reptiles. Kjoss and Litvaitis [19] demonstrated that funnel traps with drift fence arrays were more effective than thermoregulation refugia, like cover boards. However, Crosswhite et al. [20] found that visual searches outperformed drift fence arrays and funnel traps. Camera traps were considered more efficient than cage traps and thermoregulation refugia [21], while Hutchens and DePerno [22] stated that multiple techniques were best to survey terrestrial reptiles. To our knowledge, no study has compared the numerous active and passive techniques to survey STEL. Therefore, the goal of this study was to assess various reptile survey methods that effectively survey for the Plateau and Tamaulipan STEL. Our objectives were to: 1) evaluate the efficacy of various survey methods (i.e., systematic visual searches, quadrant searches, road cruising, remote cameras, pitfall traps, funnel traps, cover boards, rock mounds, detector dog surveys, and environmental DNA sampling) in determining the presence and relative abundance of STEL populations, and 2) determine if a STEL density threshold was required before STEL presence was detected.

Our hypotheses were that 1) the detector dog surveys would be the most efficient technique because trained dogs would be able to locate STEL via olfaction, even if STEL were buried, which is a natural behavior of STEL [23], or not visible by other techniques, and 2) eDNA samples will yield the most efficient means of detecting increases in STEL population density. We expected these results because 1) detector dogs performed better than other detection methods in 89% of the 542 reported cases where comparisons were made [24], and 2) eDNA would remain in the soil in detectable amounts so that as STEL populations increase, individual STEL interactions with their environment would increase, and thus, result in greater amounts of eDNA to locate, after taking into account a specific decay rate [25]. Application of eDNA has been used to determine the presence of rare species, cryptic species, and for early detection of invasive species in aquatic and terrestrial habitats [26,27].

## Materials and methods

### Study design justification

We were tasked to determine best survey methods for STEL from all known possible reptile survey and collection methods, and to determine if a threshold density of STEL was required before a survey technique would be successful. However, these tasks posed several problems. We acknowledge upfront that due to financial and logistical constraints, and permitting restrictions that a completely randomized design where all survey methods would be independent was not feasible. We identified 10 potential survey techniques. However, for complete independence and have minimal replication (N = 2 reps/treatment), 20 areas with STEL would be required. Because too few STEL populations exist and those that do are widely scattered throughout Texas [28], it was not possible to conduct surveys on wild populations. Even if the number of survey techniques was reduced, wild STEL populations would not be true replicates because of differing habitat types, climates, soils, etc. that exist throughout Texas where STEL populations currently exist. Therefore, it was decided to build a study site where such variables would be the same (i.e., apply experimental control for such variables). However, a similar problem would occur in that 20 enclosures would be required for independence of all identified techniques. Sufficient property size that was university-owned was available, but it required extensive brush removal and conversion to make the habitat suitable for STEL. Unfortunately, the cost of habitat conversion of 20 ha (20, 1-ha areas) was considered cost prohibitive by the State of Texas. In addition, to stock each enclosure with enough STEL to determine the density threshold needed for each survey technique (i.e., up to 40 STEL/ha/enclosure; Rangel [5] estimated STEL use ~270 m$^2$/STEL), it would require the capture of 800 STEL. Although the estimated wild population of STEL in Texas is unknown, we

believed such an extreme collection would cause local extirpation of remaining wild STEL populations. Also, Texas Parks and Wildlife Department only allotted our permit to capture 25 *Holbrookia lacerata* and 25 *Holbrookia subcaudalis*. Thus, to reach threshold densities of 40 STEL/ha, we needed to use both species concurrently, rather than as separate experiments. It was decided to build a single 1-ha enclosure in which to test the various survey techniques, even though several of those techniques create dependencies (e.g., a STEL could not be captured within a funnel trap and pitfall trap simultaneously; therefore, such methods would not be independent.

## Enclosure construction

Presently, there are only three known populations of *H. subcaudalis* remaining in the United States: one population in Nueces County, Texas, a second population in Jim Wells County, Texas, and the last population on Laughlin Air Force Base in Val Verde and Kinney counties, which the latter population is approximately 400 km from the other two populations [4,5]. The Nueces and Jim Wells County populations are associated with crop field margins, while the Del Rio population is associated with a sparse grassland next to the airfield on Laughlin Air Force Base.

Six isolated populations of *H. lacerata* are known to occur in central Texas; many populations along county borders of Edwards, Real, Crockett, Schleicher, Coke, Sutton, Kimble, Mason, McCullough, Tom Green, Reagan, Irion, Glasscock, Sterling, and Runnels counties [4]. Four and two populations of *H. lacerata* are associated with field margins of crop fields and short grass habitat, respectively.

Even though historically STEL were considered a grassland species [29], because the majority of current STEL populations were associated within field margins adjacent to crop fields,we built our enclosure to resemble the habitat where the majority of STEL are currently found. We removed the vegetation, plowed the soil until it was an even, pliable consistency, and erected a 1-m tall, aluminum flashing fence that was buried 30 cm deep to enclose a square 1-ha area. This was done because both species of STEL are known to be associated with plowed crop fields [11,30]. The 1-ha enclosure was delineated and numbered into 100, 10 x 10 m grids, red flagging was used to mark the corners of each grid and yellow flags were placed in the center of each grid and numbered to identify the grid (Fig 1). A driving path for an ATV was established along the outside perimeter of the fence. We allowed volunteer plants to germinate within the enclosure (e.g., Johnsongrass (*Sorghum halepense*), crabgrass (*Digitaria* spp.), Maximilian daisy (*Helianthus maximiliani*), just as such plants occurred along the margins of crop fields of where STEL were collected. We maintained the volunteer plants within the enclosure at approximately the same density, biomass, and percent cover per ha as occurred at the collection site.

To reduce the risk of predation, we maintained a predator control program throughout the study. We maintained loose soil around the outside perimeter of the enclosure and checked for predator tracks daily. We set Havahart traps baited with sardines around the perimeter of the enclosure and removed raccoon (*Procyon lotor*), opossum (*Didelphis virginiana*), striped skunk (*Mephitis mephitis*), and feral domestic cats (*Felis catus*) as they occurred in the area. We shot skyrocket fireworks into the air randomly throughout each day to scare potential avian predators from the area. Because STEL burrow underground and remain buried throughout the night [23], we considered predation by owls unlikely.

## Lizard collection and stocking

We collected 40 STEL (20 Plateau and 20 Tamaulipan STEL; 1M:1F sex ratio) during May – July, 2021, from Tom Green (31.38194 N, −100.31361 W; WGS 84) and Nueces (27.71444 N, −97.84250 W; WGS 84) counties, respectively, via road-cruising adjacent to crop fields. We maintained STEL in individual 37.8-l aquaria with 10 cm of sandy loam soil and equipped with a reptile 100-watt ceramic heat emitter bulb (ZooMed Laboratories, San Luis Obispo, CA 93401), ReptiSun 10.0 UVB, 13-watt, compact fluorescent lamp (ZooMed Laboratories, San Luis Obispo, CA 93401), and a Repti Basking 100-watt spot LED lamp (ZooMed Laboratories, San Luis Obispo, CA 93401). We maintained STEL in captivity [5] until they were randomly placed within the 1-ha enclosure.

| | | | | | | | | | |
|---|---|---|---|---|---|---|---|---|---|
| RC 1 | RC 2 | RC 3 | RC 4 | RC 5 | RC 6 | RC 7 | RC 8 | RC 9 | RC 10 |
| RC 11 | PFA 12 | VS 13 | RM 14 | 15 | CAM 16 | 17 | PFA 18 | CAM 19 | RC 20 |
| RC 21 | DD 22 | eDNA 23 | FTA 24 | 25 | 26 | 27 | eDNA 28 | VS 29 | RC 30 |
| RC 31 | CB 32 | CAM 33 | 34 | RM 35 | 36 | VS 37 | DD 38 | FTA 39 | RC 40 |
| RC 41 | RM 42 | 43 | CB 44 | 45 | 46 | CAM 47 | 48 | CB 49 | RC 50 |
| RC 51 | 52 | 53 | 54 | DD 55 | PFA 56 | 57 | 58 | 59 | RC 60 |
| RC 61 | PFA 62 | FTA 63 | CB 64 | eDNA 65 | 66 | RM 67 | VS 68 | FTA 69 | RC 70 |
| RC 71 | DD 72 | VS 73 | eDNA 74 | 75 | CAM 76 | 77 | PFA 78 | 79 | RC 80 |
| RC 81 | CB 82 | 83 | FTA 84 | 85 | RM 86 | DD 87 | eDNA 88 | 89 | RC 90 |
| RC 91 | RC 92 | RC 93 | RC 94 | RC 95 | RC 96 | RC 97 | RC 98 | RC 99 | RC 100 |

**Fig 1. Study design of the 1-ha enclosure divided into 100, 10 x 10 m grids with random placement without replacement of 9 herpetofauna collection methods (PFA=pitfall array, N=5; FTA=funnel trap arrays, N=5; RM=rock mounds, N=%; CB=cover boards, N=5; eDNA=soil samples for spot-tailed earless lizard DNA, N=5; CAM=remote cameras, N=5; DD=detector dog searches, N=10; VS=visual searches, N=10; RC=road cruising; N=4).** Collection methods were conducted 3 times with spot-tailed earless lizard densities of 5-, 10-, 20-, 30-, and 40 lizards/ha during August – September, 2001.

Lizard density in our enclosure was successively increased over a 50-day study period by adding lizards to achieve densities of 5, 10, 20, 30, or 40 lizards. At the onset of the study, STEL were randomly placed within the 1-ha enclosure at a density of 5 (3 Plateau (1M:2F): 2 Tamaulipan (1M:1F)), then 10 (5 Plateau (2M:3F): 5 Tamaulipan (3M:2F)), then 20 (10 Plateau (5M:5F): 10 Tamaulipan (5M:5F)), then 30 (15 Plateau (7M:8F): 15 Tamaulipan (8M:7F)), and finally 40 (20 Plateau (10M:10F): 20 Tamaulipan (10M:10F)) lizards for a 12 -day period at each density during August – September, 2021 (Table 1). A random number generator from 1–100 was used with replacement to select a quadrant to place individual STEL, which were released at the center point of their randomly selected quadrant. Lizards were provided 6 days to acclimatize to the enclosure, then each survey method was conducted three times during a 6-day period, and STEL species and number observed were recorded (Fig 2). Coloration was used to identify the species of STEL because Plateau STEL are a caramel tan color while Tamaulipan STEL are a slate grey color [3,12] (S1 Photo).

Because we placed a total of 5, 10, 20, 30, and 40 STEL into a 1-ha, outdoor enclosure, we believe the actual STEL density was known during the study. The study was conducted during a 60-day period; therefore, additional STEL due to reproduction within the enclosure was not possible because STEL egg incubation is approximately 5 weeks [12,31]. Also, the enclosure walls were not conducive for STEL emigration, and although our study area was located within the historic range of Tamaulipan STEL, none have been documented at the study location for at least 3 decades (SE Henke, pers. observ.), which made immigration unlikely. Lastly, the briefness of the study and our predator control efforts reduced the likelihood of STEL loss due to depredation and STEL carcasses were not found that would indicate mortality by other causes.

At the completion of this study, STEL were maintained within the enclosure to continue the investigation into the potential success of translocation. Currently, STEL survival, reproduction, and longevity are unknown following translocation.

**Table 1. Number of male and female Plateau (*Holbrookia lacerata*) and Tamaulipan (*H. subcaudalis*) spot-tailed earless lizards (STEL) placed into the 1-ha enclosure to determine if a density threshold was needed before STEL became detectable in southern Texas during August – September 2021.**

| Density (1 ha) | | Plateau | | Tamaulipan | | | Overall sex ratio |
|---|---|---|---|---|---|---|---|
| | | Male | Female | Male | Female | | |
| 5 | | 1 | | 1 | 1 | | 2M:3F |
| | *Add* | (+1) | | (+2) | (+1) | | |
| 10 | | 2 | 2 | 3 | 2 | | 5M:5F |
| | *Add* | (+3) | (+1) | (+2) | (+3) | | |
| 20 | | 5 | 3 | 5 | 5 | | 10M:10F |
| | *Add* | (+2) | (+2) | (+3) | (+2) | | |
| 30 | | 7 | 5 | 8 | 7 | | 15M:15F |
| | *Add* | (+3) | (+3) | (+2) | (+3) | | |
| 40 | | 10 | 8 | 10 | 10 | | 20M:20F |

## Survey techniques

At each lizard density, STEL were assessed by 10 commonly used survey methods according to Dodd [32]. We conducted pitfall trapping [33], funnel trapping [34], searches of rock mounds [15] and cover boards [35,36], remote camera surveys [37], systematic visual surveys [38], quadrant searches [39], road cruising [40,41], detector dog searches [42], and eDNA samples [43] within the 1-ha enclosure. Each collection method was placed in the center of randomly selected grids within the inner 64 grids (i.e., 8 x 8) without replacement and replicated five times (Fig 1). The outer 36 grids were designated for road cruising (Fig 1), which was purposely selected so STEL could be visible from a vehicle without the vehicle driving inside the enclosure and potentially striking a STEL.

**Pitfall traps.** Pitfall traps (N = 5) consisted of 4, 5-liter buckets buried so the lip of the bucket was level with the ground and placed in an even-spaced triangular pattern with a central bucket. Buckets were separated by 5-m and silt fencing was partially buried upright to make a drift fence that led to each bucket. Bucket lids were removed when trapping occurred and elevated about 5 cm above the bucket with wooden sticks to provide shade inside the bucket. Lids were secured on buckets when trapping was complete for the sampling period. Buckets were checked every 8-hours when in use and captured fauna were enumerated and recorded during each 24-hour period.

**Funnel traps.** Cylinder-shaped funnel traps, 1-m in length, were made of screening material that had a 30-cm opening at one end and closed at the other end. Funnel trap arrays (N = 5) consisted of a 10-m long silt fence that was partially buried upright to make a drift fence that led to a funnel trap at each end of both sides of the drift fence. Funnel traps were checked every 8-hours when in use and captured fauna were enumerated and recorded during each 24-hour period. Funnel traps were removed when not in use.

**Rock mounds.** Rock mounds (N = 5) were constructed of 14, 10 × 10 × 4 cm patio bricks piled on top of each other in a pyramid fashion. This structure allowed STEL a natural-looking hiding cover and a thermoregulation structure for basking and cooling. Rock mounds were searched by hand for STEL during surveys and STEL were enumerated and recorded. Rock mounds were reassembled after each search.

**Cover boards.** Plywood cover boards (N = 5), measuring 1.2 m × 1.2 m × 1.25 cm, were placed on top of 4 patio bricks, previously described, at each corner of the plywood board so the board rested about 4 cm above the ground surface. Like the rock mounds, this structure allowed STEL hiding cover and a thermoregulation structure for basking and cooling. Capture net was placed at the back side of cover boards as they were lifted away from the searcher to entice hidden STEL to flee toward the direction of the net. Cover boards were returned to their original placement once checked.

**Remote cameras.** Sawhorses (N = 5), measuring 2.5 m long and 3 m tall, were built from 5 × 10-cm wood and equipped with 2 Reconyx remote-triggered Hyperfire 2 Covert IR game cameras (Reconyx, Holmen, WI 54636) mounted

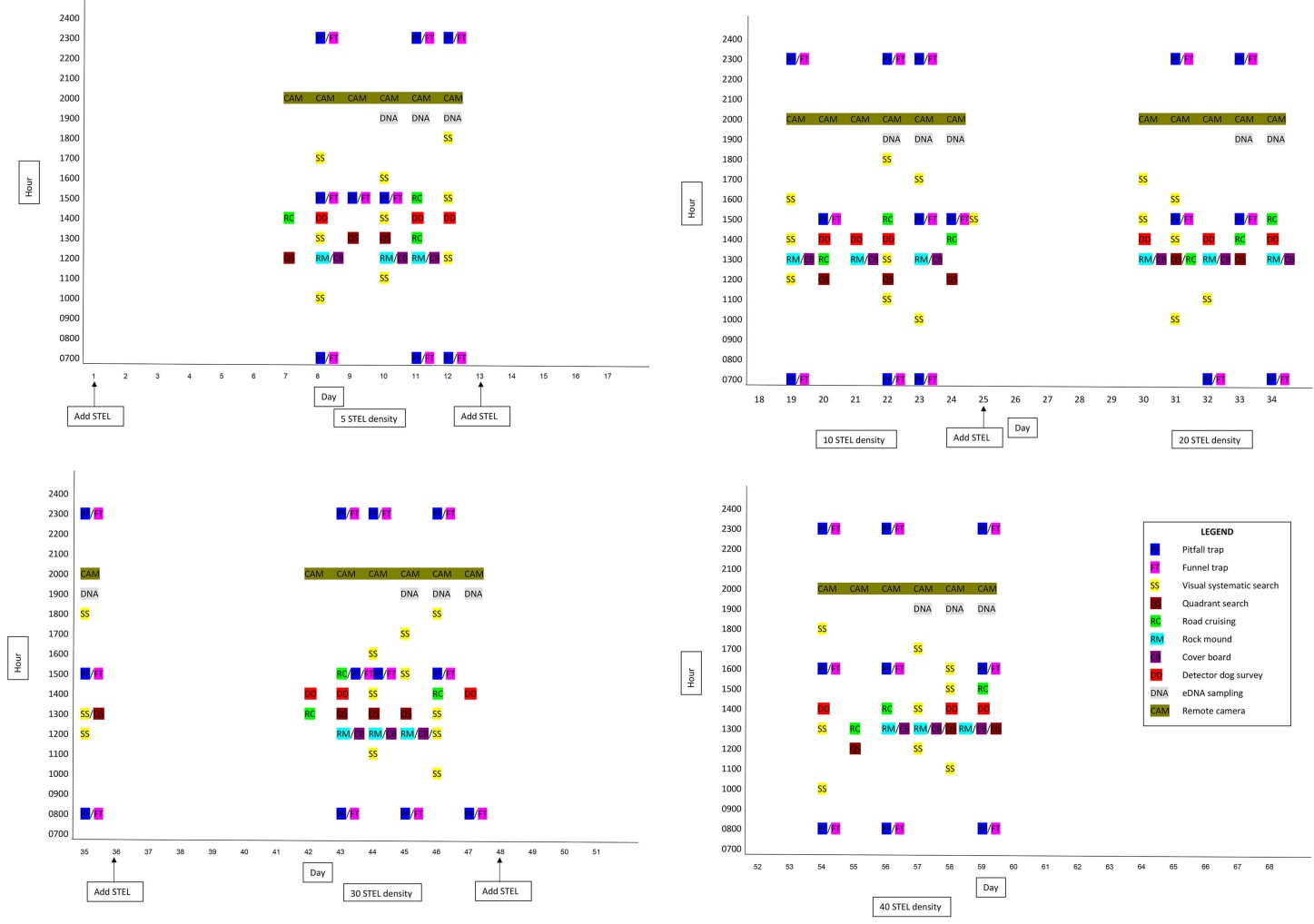

**Fig 2. Chronology of spot-tailed earless lizard placement and survey method schedule during August – September, 2021, within the 1 ha enclosure.**

at each end that photographically recorded species directly beneath the cameras that activated the motion sensor triggers of the camera. Cameras were placed viewing straight down, suspended from sawhorses at a height that provided a 7.7-m² image area. Cameras were turned on at the beginning of each survey period and recorded images continuously until the end of the survey period. Images were recorded to species.

**Quadrant searches.** Five grids were randomly chosen for each day of the 3-day survey during each STEL density. Quadrants were thoroughly searched by 3 people for 20 minutes per 10 × 10 m grid. Therefore, quadrant searches were approximately 5 person-hours per survey. STEL encountered within the quadrant were enumerated and recorded.

**Detector dog surveys.** Two detection dogs (i.e., 5-year-old female Jack Russel terriers; Coastal Bend Canine College, Kingsville, Texas 78363) were trained to locate STEL odor [42]. A live-specimen STEL was used for dog training purposes. Trained dogs, one at a time, individually searched randomly selected grids (N = 5) as directed by the trainer for 20 minutes per 10 × 10 m grid. Dogs were alternated between grids so it could rest and drink between searches. If no

STEL were found, the dogs then were allowed to search the entire 1 ha enclosure and signal if it located a STEL. Number of STEL encountered were recorded.

**Road cruising.**  Road cruising with an ATV was conducted by a driver and observer along the outside perimeter of the enclosure. This allowed for an unobstructed view of the outer 36 quadrants of the enclosure. The quantity of road measured ~400 m, which was driven <5 kph. The observer kept watch along the path for STEL and for movement, and recorded STEL observed.

**Environmental DNA sampling.**  Soil samples (25 g each) from the soil surface were collected from 10 random points within a randomly selected grid and composited as a single 250-g sample. The center flag within the randomly selected grid was used as the central point to obtain collection points. A random number generator from 0 to 8 selected the direction from the center flag for one of eight cardinal directions (i.e., N, NE, E, SE, S, SW, W, and NW), and the next number from 0–500 selected the distance (cm) from the central point to the randomly selected point. Approximately 25-g of soil from the surface at each of 10 randomly selected points was collected with a plastic spoon and soil was placed in a freezer bag. Spoons were discarded after a single use. Freezer bags were marked for quadrant and STEL density and placed in a −70 C freezer.

All samples were shipped frozen to the Helbing Lab of the University of Victoria (Victoria, British Columbia, Canada V8P 5C2) for eDNA analyses. Environmental DNA analyses were conducted according to the methods of Veldhoen et al. [44] and Langlois et al. [45] for presence of DNA matching *Holbrookia* species.

**Systematic visual searches.**  The entire 1-ha enclosure was searched by 5 observers for 60 minutes for a total of 5 person-hours. Observers were spaced 5 m apart (25 m width) and walked the length of the enclosure in 4 swaths counting STEL until the entire enclosure was searched. When a STEL was encountered, observers stopped moving to determine where the STEL would relocate as a means to avoid recounting the STEL multiple times during a single survey.

## STEL detectability

To determine STEL detectability, we conducted visual systematic surveys with 5 searchers of the entire 1-ha enclosure. Searches required ~5 person-hours to search the 1-ha enclosure. Searches were conducted as previously described in *Systematic visual searches*. We conducted searches during three time-intervals (i.e., 1000–1200 h, 1300–1500 h, 1600–1800 h) during the same day for three consecutive days at each STEL density. The number of STEL observed during each survey was recorded and the probability of detection ± SE were calculated.

**Experimental design considerations.**  Because of the logistical limitations previously described, spatial replication was not feasible and some methods were dependent on the success of other methods. Therefore, dependent survey methods (i.e., pitfall traps, funnel traps, cover boards, and rock mounds) are reported separately and their relative success described rather than statistically analyzed. Non-successful survey methods, determined post hoc, included remote camera surveys, detector dog surveys, quadrant searches, and eDNA sampling. The non-successful methods also are reported separately and their relative success described rather than statistically analyzed. Thus, two sets of hypotheses were tested: *Hypothesis Set One* tested the effect of survey methods (i.e., road cruising or systematic visual searches), STEL density, and their interaction on observed number of STEL; *Hypothesis Set Two* tested the effect of time-of-day of systematic visual searches, STEL density, and their interaction on observed number of STEL. For Hypothesis Set One, the enclosure was surveyed over a 6-day period at each STEL density; during each 6-day period, we randomly assigned 3 of these survey days for road cruising and 3 (different) days for systematic visual searches. For Hypothesis Set Two, the enclosure was surveyed over a 6-day period using systematic visual searches at 3 times of day (1000–1200 h; 1300–1500 h; and 1600–1800 h), with 3 survey days randomly selected for each time of day.

## Statistical considerations

**Pseudo-replication and independence.**  Because of lack of spatial replication, we surveyed over time, with 3 survey days for each sampling method (Hypothesis Set One) and 3 survey days for each time of day (Hypothesis Set Two).

Survey days are, therefore, (temporal) pseudo-replicates of a single enclosure, and lack of independence over survey days might be anticipated. Following Christiansen and Bedrick [46], we tested for independence among survey days at each STEL density. For Hypothesis Set One, the hypothesis of independence was not rejected for densities ranging from 5 to 40 STEL ($P = 0.75, 0.75, 0.25, 0.5728$, and $0.5242$, respectively); similar results were observed for Hypothesis Set Two ($P = 0.44, 0.44, 0.16, 0.7901$, and $1.00$). Based on these results, we considered the 3 survey days for each Hypothesis Set as independent pseudo-replicates, and we limit statistical inference to this enclosure [47].

**Statistical model details for each hypothesis set.** For Hypothesis Set One, we used a linear mixed model with sampling method and STEL density, as well as their interaction, as fixed effects and survey day nested within sampling method as a random effect. Density of STEL was modeled as a repeated measures factor to accommodate possible lack of independence (because increased STEL density was implemented over a 50-day study period). We modeled a variety of variance-covariance structures for the repeated measures aspect of our analysis to account for the possibility of non-independence (because of repeated measures) and heteroscedasticity [48]: first-order autoregressive, compound symmetry, and Toeplitz (and their heteroscedastic versions), first order autoregressive-moving average, variance-components, and unstructured. Patterns of non-independence need not have a subject-matter justification [49]; their use is to account for the lack of independence common in repeated measures and longitudinal data sets, regardless of the reasons that are responsible for the lack of independence so that appropriate standard errors can be estimated and valid inferences can be drawn. Additionally, accounting for lack of independence generally increases the precision with which model parameters can be estimated [48].

In addition, to test how STEL detection probability varies as a function of sampling method and lizard density (Hypothesis 1), we used generalized linear (mixed) models (GLMs/GLMMs). We modeled the number of detections using a binomial distribution, summing STEL detections across the three survey repetitions per plot. Sampling method (visual searches and road cruising), lizard density, and their interaction were included as fixed effects. In addition, plot identity nested within sampling method was included as a random effect to account for potential dependence structures in the experimental design. The necessity of including random effects was assessed post hoc by inspecting the variance components associated with the random terms [50]. Because some levels of the fixed factors exhibited perfect separation (i.e., zero detections in all cases of a level), we fitted the models using a Bayesian approach. Bayesian modeling provides a robust framework for handling perfect separation and small sample sizes, as the inclusion of priors enables stable estimation of posterior distributions [51]. Model comparisons were performed using leave-one-out cross-validation (LOO), where higher LOO values indicate better predictive performance [52,53].

For Hypothesis Set Two, systematic visual searches were conducted at 3 times of day (1000–1200 h; 1300–1500 h; and 1600–1800 h) throughout each survey day, providing an opportunity to assess time-of-day effects on STEL detection for this survey method. This analysis was similar to the analysis described above for the comparison between road cruising and systematic visual searches, with the following differences: (1) when comparing road cruising and systematic visual searches (Hypothesis Set One), the factors of interest were sampling method and STEL density, and data from only time-of-day (i.e., 1300–1500 h) were used; (2) to assess time-of-day effects for systematic visual searches (Hypothesis Set Two), the factors of interest were time of day (1000–1200 h; 1300–1500 h; and 1600–1800 h) and density. Therefore, the factor of sampling method in Hypothesis Set One was replaced with the factor of time-of-day in Hypothesis Set Two; other aspects of the analyses were similar.

All models for the probability of detection were fitted using the brms package [54] in R [55], with 6000 iterations, 10 thinning intervals for hypothesis 1 models and 50000 iterations, 25 thinning intervals for hypothesis 2 models. Four Markov chains, 1000 warmup steps, and uninformative priors were used for all models. Model convergence was assessed via the Gelman-Rubin statistic (R), where R values close to 1 indicate good mixing [56], and was visually inspected using trace plots. Finally, model fit and assumptions related to dispersion were evaluated using posterior predictive checks and residual diagnostics implemented via the DHARMa [57] and DHARMa.helpers packages for Bayesian models [58].

**Response variable distribution.** Our response variable for the above analyses, number of observed STEL, was a count datum. Traditional methods of analyzing count data with logarithmic or square root transformations have been criticized [59,60] in favor of generalized linear mixed model with appropriate link functions to accommodate common count-related distributions (e.g., negative binomial and Poisson). Others (e.g., [61–63] have reported that $P$ values based on log-transformed data and on generalized mixed models often lead to similar conclusions. In addition to dependent variable transformation or use of generalized linear models, a third alternative is available: a linear mixed model using permutation-based methods [64] to test hypotheses.

Given the large number of zeroes (depending on STEL density) in our count data, we employed 3 analytical strategies for each Hypothesis Set, and some preliminary results are presented here to justify our final analytical approach. *Strategy A:* linear mixed models (PROC MIXED in SAS) with dependent variable transformation. We analyzed STEL, the number of observed spot-tailed earless lizards; the square root of STEL (hereafter, SSTEL); the log(STEL + 1), hereafter, LSTEL; and a normal-score transformation [65,66] of STEL (hereafter. NSTEL). *Strategy B:* generalized linear mixed models (PROC GLIMMIX in SAS) with link functions appropriate for STEL as a negative binomial and as a Poisson distributed random variable. *Strategy C:* permutation-based analysis of variance (PERMANOVA in PRIMER-E +) with the same linear mixed model.

We encountered convergence problems using generalized linear mixed models (Strategy B). Relative to Strategy A and Hypothesis Set One, conclusions about main effects were similar for all 4 scales of measurement, and although the test of the interaction was not significant at any scale, the $P$ value for STEL ($P = 0.3118$) was lower than $P$ values on the various transformed scales, which varied from $P = 0.5148$ (SSTEL), $P = 0.5941$ (NSTEL), and $P = 0.6108$ (LSTEL). For Strategy A and Hypothesis Set Two, conclusions for main effects were similar regardless of scale of measurement; however, $P$ values increased ($P = 0.0057, 0.0545, 0.1173$, and $0.1991$) for STEL, SSTEL, LSTEL and NSTEL, respectively. Permutation-based $P$ values (Strategy C), however, were generally similar to those estimated when STEL was analyzed using Strategy A both for main effect and interaction tests.

Thus, with respect to tests of main effects, our results support conclusions of Ives [61], Stroup [62], St-Pierre et al. [63]: $P$ values led to similar conclusions for all 4 scales of measurement. This was not the case, however, for the interaction test for either Hypothesis Set—and for this test, permutation-based $P$ values were more similar to Strategy A values on STEL than to $P$ values on transformed data (which varied considerably depending on scale of measurement), including the normal score transformation that is recommended for interaction tests [65]. Based on these results, we chose permutation-based methods to test hypotheses.

Our research protocols were approved by an authorized animal care committee at Texas A&M University-Kingsville (#05272021/1469), and we have followed the code of practice adopted for the reported experimentation and methodology.

## Results

### Comparison of dependent survey techniques and non-successful survey techniques

Of the 10 survey methods attempted, funnel traps, rock mounds, cover boards, remote cameras, and detector dogs yielded no STEL at any density. Pitfall trapping captured 1 Plateau STEL, but fire ants (*Solenopsis invicta*) killed the STEL before our next check. Quadrant searches and eDNA sampling yielded two Tamaulipan STEL and one STEL (species not discernable), respectively, but only at a density of 40 STEL/ha. Remote cameras took 65,317 photographs, of which 2,763 photos (4.2%) contained white-tailed deer (*Odocoileus virginianus*), 347 (0.5%) contained insects (e.g., walking sticks (*Diapheromera femorata*), grasshoppers, (Order: Orthropoda), crickets (Order: Orthoptera), moths (Order: Lepidoptera), etc.), 222 (0.34%) contained small mammals (e.g., cotton rats (*Sigmodon hispidus*) and cottontail rabbits (*Sylvilagus floridanus*)), while the remaining photographs (i.e., 61,985; ~95%) were of wind-blown vegetation and resulting shadows

caused by the sun's angle on the camera mount. STEL were not observed in any photograph. Only systematic visual searches of the entire enclosure and road cruising produced quantifiable results.

## Comparison of road cruising and systematic visual searches

On average, systematic visual searches observed nearly twice as many STEL (1.5±0.2 STEL/survey) than the number of STEL observed during road cruising (0.8±0.2 STEL/survey; Fig 3). Survey method (road cruising or systematic visual search) and STEL density acted independently ($F_{4, 16}$=1.24, $P$=0.302) in their effects on observed STEL density (Fig 3). More STEL were observed ($F_{1,4}$=7.56, P=0.05) with systematic visual searches than with road cruising. Additionally, the number of observed STEL varied ($F_{4, 16}$=6.9, $P$=0.004) among STEL densities, regardless of survey method, generally increasing as STEL density increased (range=0.3±0.3 STEL/survey at 5 STEL ha$^{-1}$–2.3±0.3 STEL/survey at 40 STEL ha$^{-1}$; Fig 3).

We found that road cruising yielded higher detection probabilities for STEL compared to visual systematic searches, with estimated odds of detection being consistently higher across lizard densities (β=2.06, 95% CPD: –0.15 to 4.37; Table 2; Fig 4). When comparing detection probability across densities within visual searches, detection odds were significantly higher at 40 STEL/ha compared to 10 STEL/ha (odds ratio=0.06 95% CI: 0.0005–0.36), 20 STEL/ha (0.045 0.00003–0.46), and 30 STEL/ha (0.049 0.00009–0.57), suggesting improved detectability with higher densities (Table 3). However, comparisons between low to intermediate densities (e.g., 10 vs. 20 or 30 STEL/ha) showed high uncertainty, with Bayesian credible intervals crossing 0, indicating no clear difference in detection probability. For road cruising, a similar pattern was observed: detection odds increased with density, particularly between the lowest and highest levels. The odds of detecting STEL at 40 STEL/ha were over 11 times higher than at 5 STEL/ha (odds ratio=11.64, 95% CI: 0.41–153.06), though with wide uncertainty (Table 3). As with visual searches, the contrasts among intermediate densities had overlapping CI intervals with 0, suggesting non-significant pairwise differences. Posterior estimates from the Bayesian model showed adequate convergence (all Rhat≈1.00, ESS>1700), and residual checks indicated no violation of distributional assumptions (see Table 2 and Supplementary material 1).

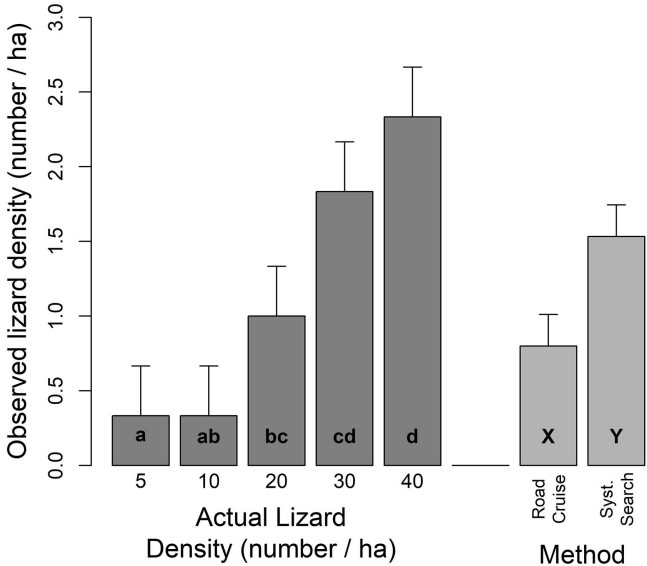

**Fig 3. Observed lizard density (ha$^{-1}$) detected at actual lizard densities of 5 to 40 lizards ha$^{-1}$, and by either road cruising or systematic searches.** Density means with the same lower case letter (a, b) are not significantly different; survey method means with the same upper case letter (X, Y) are not significantly different (protected LSD test, P>0.05).

**Table 2. Posterior estimates of model parameters evaluating the effect of sampling method (Visual Systematic Searches vs. Road Cruising) and animal density on detection probability.**

| Coefficient | Estimate | SE | Lower 90% CI | Upper 90% CI | Rhat | ESS |
|---|---|---|---|---|---|---|
| Visual systematic searches | −4.72 | 1.14 | −7.1 | −2.72 | 1 | 1828.6 |
| Density20 | −0.33 | 1.54 | −3.57 | 2.5 | 1.002 | 1919 |
| Density30 | −0.17 | 1.52 | −3.29 | 2.77 | 0.999 | 2032.7 |
| Density40 | 2.92 | 1.19 | 0.7 | 5.32 | 1.003 | 1781.8 |
| Density5 | −0.67 | 1.63 | −3.93 | 2.41 | 1 | 2045.6 |
| Road cruising | 2.06 | 1.14 | −0.15 | 4.37 | 1 | 1918.3 |
| Density20: Road cruising | 1.09 | 1.65 | −2.23 | 4.25 | 1.001 | 1964.5 |
| Density30: Road cruising | 1.49 | 1.57 | −1.39 | 4.75 | 1 | 2095.8 |
| Density40: Road cruising | −0.58 | 1.28 | −3.06 | 1.9 | 1 | 1923.5 |
| Density5: Road cruising | 0.41 | 1.76 | −2.92 | 3.84 | 1 | 2140.5 |

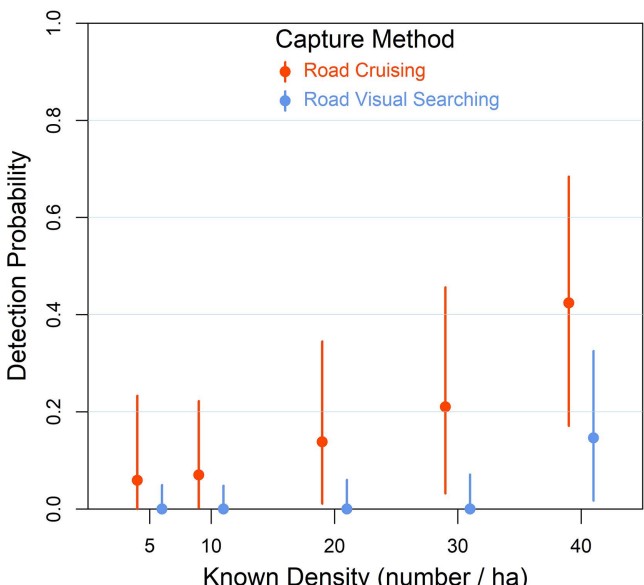

**Fig 4. Posterior predictive estimates of detection probability of STEL as a function of capture method and known density of lizards with 90% Bayesian credible intervals.**

## Effect of time of day during systematic searches

Time of day and STEL density interacted ($F_{8,24} = 3.74$, $P = 0.005$) in their effects on observed STEL (Fig 5). Time of day effects were detected only at lizard densities of 30 ha$^{-1}$ ($F_{2, 6} = 16.3$, $P = 0.004$) and 40 ha$^{-1}$ ($F_{2, 6} = 11.4$, $P = 0.006$) where more STEL were observed at 1300–1500 h than either at 1000–1200 h or 1300–1500 h. Density effects were detected ($F_{4, 8} = 7.73$, $P = 0.009$) only during the 1300–1500 h time period when more STEL were observed at densities of 30- and 40 ha$^{-1}$ than at densities of 5, 10, and 20 ha$^{-1}$ (Fig 5).

## STEL detectability

Detectability of STEL, based on known STEL density, ranged from 3.3–8.9%, with a mean of 6.6 ± 0.9% (Table 4). Detectability rates did not improve as STEL density increased.

**Table 3. Pairwise contrasts of detection probabilities across animal density levels, by sampling method.**

| Contrast | Treatment | Odds Ratio | Lower 90% CI | Upper 90% CI |
|---|---|---|---|---|
| DENSITY10 / DENSITY20 | Visual Systematic Search | 1.32 | 0.01 | 20.02 |
| DENSITY10 / DENSITY30 | Visual Systematic Search | 1.13 | 0.01 | 14.87 |
| DENSITY10 / DENSITY40 | Visual Systematic Search | 0.06 | 0 | 0.36 |
| DENSITY10 / DENSITY5 | Visual Systematic Search | 1.84 | 0.02 | 32.85 |
| DENSITY20 / DENSITY30 | Visual Systematic Search | 0.88 | 0 | 21.62 |
| DENSITY20 / DENSITY40 | Visual Systematic Search | 0.05 | 0 | 0.46 |
| DENSITY20 / DENSITY5 | Visual Systematic Search | 1.44 | 0 | 43.64 |
| DENSITY30 / DENSITY40 | Visual Systematic Search | 0.05 | 0 | 0.57 |
| DENSITY30 / DENSITY5 | Visual Systematic Search | 1.61 | 0 | 54.04 |
| DENSITY40 / DENSITY5 | Visual Systematic Search | 33.72 | 0.25 | 707.88 |
| DENSITY10 / DENSITY20 | Road Cruising | 0.48 | 0.01 | 3.4 |
| DENSITY10 / DENSITY30 | Road Cruising | 0.28 | 0.01 | 1.68 |
| DENSITY10 / DENSITY40 | Road Cruising | 0.1 | 0 | 0.55 |
| DENSITY10 / DENSITY5 | Road Cruising | 1.22 | 0.01 | 14.79 |
| DENSITY20 / DENSITY30 | Road Cruising | 0.57 | 0.01 | 3.06 |
| DENSITY20 / DENSITY40 | Road Cruising | 0.21 | 0.01 | 1.02 |
| DENSITY20 / DENSITY5 | Road Cruising | 2.51 | 0.06 | 35.54 |
| DENSITY30 / DENSITY40 | Road Cruising | 0.37 | 0.02 | 1.69 |
| DENSITY30 / DENSITY5 | Road Cruising | 4.42 | 0.09 | 54.98 |
| DENSITY40 / DENSITY5 | Road Cruising | 11.64 | 0.41 | 153.06 |

## Discussion

We determined that the best method to observe the most STEL was systematic visual searches; however, systematic visual searches require greater time to survey an area, especially if the presence of STEL within the area is unknown. Therefore, road cruising may be the optimum survey choice to first determine the presence of STEL in an area, and then switch to systematic visual searches to enumerate the population of STEL. Road cruising has the advantages of traversing a greater distance than walking within the same allotment of time, and surveyors can get closer to STEL with vehicles than approaching them on foot, which aids species verification [11,67]. We acknowledge that it was possible that STEL behavior could have been altered by the translocation process. Cooper [68] noted that lizards from the *Holbrookia* genus changed behavior due to autotomy. Therefore, it is possible that STEL became wary from translocation and thus became less active, which would decrease detectability. However, we noted that upon direct approach during searches within the enclosure, STEL would allow a similar approach distance before fleeing, as they did in their original habitat prior to collection [67]. Thus, we believe translocation did not alter STEL normal behavior. In addition, we acknowledge that ground vibrations created by vehicles can alter reptile movements, with some species fleeing from vehicles while other species seek shelter as vehicles approach [69,70]. However, as a survey technique for STEL, road cruising did yield higher detection probabilities than the other survey methods. Thus, it appears STEL may exhibit the former rather than latter behavior.

Also, we acknowledge that STEL were placed into an enclosure area that was modified from its natural vegetative state. However, even though STEL historically were considered a grassland species [30], STEL currently seem to prefer early successional, heavily disturbed areas [11]. LaDuc et al. [4] and Rangel [5] noted the relative abundance of STEL was greatest along the periphery of crop fields. Thus, we converted southern Texas shrubland into a heavily disturbed plowed field with volunteer plants to resemble the habitat in which STEL were originally found. We believed the translocation to such habitat was a less dramatic change to STEL and thus, less likely to alter their behavior.

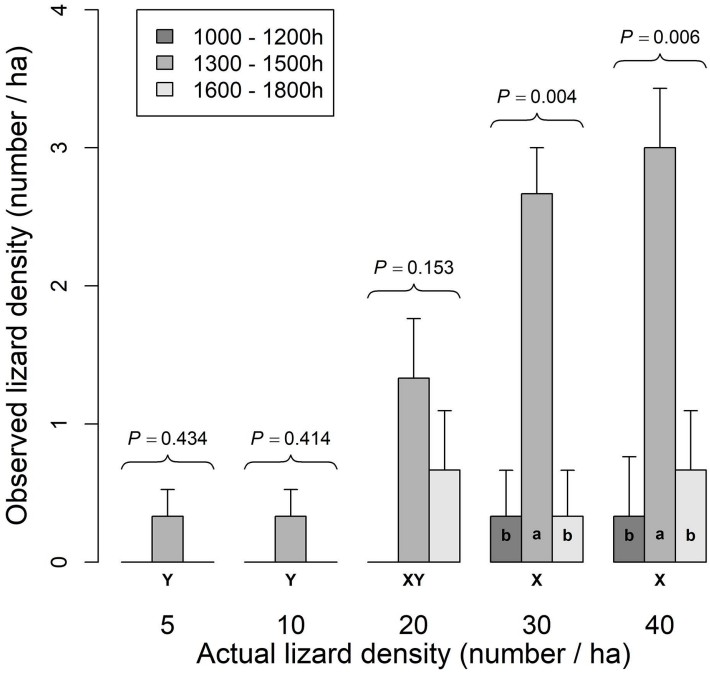

**Fig 5. Observed lizard density (ha⁻¹) detected as a function of time of day (1000–1200 h, 1300–1500 h, or 1600–1800 h) and actual lizard density (ha⁻¹).** Suspended *P* values correspond with time-of-day comparisons at given actual lizard densities; time-of-day means at a given actual lizard density with the same lower case letter (a, b) are not significantly different. Density means for the 1300–1500 h time period with the same upper case letter (X, Y) are not significantly different (protected LSD test, P > 0.05).

**Table 4. Detection rates of spot-tailed earless lizards (STEL; *Holbrookia lacerata* and *H. subcaudalis*) in a 1-ha enclosure in southern Texas during August – September 2021.**

| Actual STEL density (ha)[1] | 5 | 10 | 20 | 30 | 40 |
|---|---|---|---|---|---|
| Detectable STEL density (42%)[2] | 2.1 | 4.2 | 8.4 | 12.6 | 16.8 |
| Observable STEL density[3] | 2 | 4 | 8 | 12 | 16 |
| Visually observed STEL/survey[4] | 0.33 | 0.33 | 1.33 | 2.67 | 3 |
| Detection rates of actual density (%)[5] | 6.6 | 3.3 | 6.6 | 8.9 | 7.5 |
| Detection rate of observable density (%)[6] | 16.5 | 8.2 | 16.6 | 22.2 | 18.8 |

[1]Actual density was the known number of STEL placed within the enclosure, first at 5, then 10, then 20, then 30, and finally at 40 STEL ha⁻¹.

[2]Due to STEL burying behavior, at best only 42% of STEL were above-ground at any point in time during the diel cycle (Rangel 2023), which was defined as detectable STEL density.

[3]Because it was not possible to locate a partial STEL, we rounded the detectable STEL density down to provide a conservative estimate and defined this as the observable STEL density.

[4]Visually observed STEL was the average number of STEL observed during a visual systematic search (N = 3).

[5]Detection rate of actual density was defined as the visually observed STEL/survey divided by the actual STEL density ha⁻¹.

[6]Detection rate of observable density was defined as the visually observed STEL/survey divided by the observable STEL density.

Many survey methods and techniques we attempted were unsuccessful. Perhaps STEL within our study did not traverse the enclosure as much as expected. Hibbitts et al. [11] stated that STEL had a much larger home range (e.g., 2–7 ha) than other lizard species of similar life history characteristics. However, we noted that most STEL were fairly sedentary and typically remained within a 3-quadrant area (i.e., 300 m²; SE Henke, *pers. observ.*). Therefore, if STEL did not

move from quadrant to quadrant, then perhaps they did not encounter the various structures and drift fences, which would account for their lack of presence by many of the attempted techniques.

Such low capture rates by pitfall and funnel trapping did not warrant the effort required for construction and operation, especially if in clay soils. Clay soils require much digging effort to place buckets and drift fencing into the ground because clay soils harden significantly when dry and become quite heavy when wet [71]. Additionally, the use of trapping methods in areas where fire ants are present is not advisable because STEL ability to escape from ants is limited, and as we found, resulted in mortality.

Much of the currently known distribution of STEL occurs in association with crop fields [30]; therefore, we built our enclosure to have characteristics consistent of crop fields (i.e., plowed, pliable soil for ease of burying, volunteer forbs and grasses to serve as cover and to entice insects as a food source). However, by doing so, perhaps the need for artificial structures were not needed. STEL have been noted to bury themselves throughout a diel cycle (5,23], which we provided ample habitat in which to bury. Although STEL have been observed to use rocks for basking and crevices for hiding and/or thermoregulation, those observations were in areas of hard-compacted soils and abandoned airfields [1].

It was surprising that STEL were not observed by remote cameras. Approximately 1% of our enclosure was monitored by cameras and >65,000 photographs were taken, of which even quite small subjects such as crickets and grasshoppers were clearly visible. STEL have a documented home range of about 2.2 ha and 7.7 ha (Minimum Convex Polygon estimator) for Plateau and Tamaulipan STEL, respectively, with little overlap between conspecifics [11]. With our final density of 40 STEL within our 1 ha enclosure potentially requiring an area up to 44-154X the size of our enclosure area, it does seem reasonable that at least one STEL would have passed within the camera range. Perhaps translocation to the 1 ha enclosure altered STEL behavior to become more sedentary than described by Hibbitts et al. [11]; however, we did note that STEL did not remain within their randomly assigned quadrant of original placement within the enclosure. STEL would relocate to sometimes distant quadrants from their original placement, but once acclimated to the enclosure, remained within a 300 m$^2$ area.

Although detector dogs have experienced much success with other reptile species such as eastern box turtles (*Terrapene carolina carolina*) [72], forest geckos (*Hoplodactylus granulatus*) [73], and brown tree snakes (*Boiga irregularis*) [74], detector dogs in our study were unable to locate STEL. Though the two dogs each appeared highly motivated and kept their noses to the ground during surveys, and both dogs were able to successfully locate STEL from numerous containers during training, none ever signaled that a STEL was located. We acknowledge that the issue may have been with the dogs that were trained; however, we equally surmise that perhaps STEL do not produce a scent, or possibly they produce a substance that can hide their scent from potential predators (i.e., chemical camouflage). West African savannah frogs (*Phrynomantis microps*) release skin secretions that prevent the aggressive stinging behavior of ponerine ants (*Paltothyreus tarsatus*) so the frogs can live unharmed among the ants [75]. We did note that after handling STEL, an obvious scent on our hands was not apparent as is the case if handling other reptile species such as a checkered garter snake (*Thamnophis marcianus*) or Bosk's fringe-toed lizard (*Acanthodactylus boskianus*) [76]. However, this is only speculative and needs further study.

Environmental DNA sampling is an emerging tool that has promise as a survey technique for STEL. However, at present, we equate the method to 'finding a needle in the haystack.' One must collect a soil sample from upon which a STEL rested. Close to an actual resting spot will still produce a negative result. Our hypothesis was that as STEL density increases and STEL transverse the 1 ha enclosure, the greater the probability of locating *Holbrookia* DNA. However, high humidity combined with warm temperatures, common conditions in southern Texas, may increase the activity of DNA-scavenging microorganisms, which can result in template degradation [77]. In addition, UV light can increase the formation of thymine dimers that render the DNA unsuitable for qPCR analysis [78]. Thus, the more time that STEL were in the enclosure did not equate to greater probability of finding *Holbrookia* DNA because of climatic factors (i.e., high temperature, high humidity, increased UV intensity). Also, Adams et al. [79] developed the Shedding Hypothesis, which states that keratinized skin, as found in reptiles, may shed less DNA than species with semipermeable skin. Thus, eDNA appears

well-suited for aquatic environments [26], but the technology has complications with terrestrial habitats and species. For example, eDNA was successful in locating Burmese python (*Python bivittatus*) refugia, but only with telemetry-monitored pythons. The method was not successful in locating snake refugia without the knowledge of known sites. We believe the method could potentially be successful if an attractant could be developed to entice STEL to a specific site. Such has been the case in attracting sharp-tailed snakes (*Contia tenuis*) to use asphalt shingles for thermoregulation and then swabbing the artificial cover object for eDNA detection [27].

It is worthy to note that even with the knowledge of the actual density of STEL, one will, at best, observe less than half (i.e., 42%) because of their burying behavior [9]. Unfortunately, even with this knowledge, STEL density was not predictable and even an assessment of relative abundance between areas could be problematic. At low densities, the number of observable STEL did not differ. Even as densities became greater, observable STEL numbers did not increase substantially, or in proportion to their increasing density. Therefore, observing STEL in an area only reliably allows one to note their presence. As with many rare and elusive species, not finding them in an area does not necessarily equate to their absence from that area, but only that they were not observed on a specific date and time.

## Conclusions

Our study provides insight into best survey methods for these species-of-concern, and for terrestrial reptiles in general. We recommend road cruising surveys as the most time-efficient method to determine STEL presence. Once a population is located, then systematic visual searches yield the greatest number of STEL per effort. Researchers of STEL who wish to enumerate STEL populations should conduct surveys between 1300–1500 hrs to yield the greatest number of STEL. Unfortunately, because of their elusive behavior, only presence/absence data should be considered. Multiple surveys conducted during multiple consecutive years appear to be needed before considering STEL absent from a given location.

## Ethics statement

Our research protocols were approved by an authorized animal care committee at Texas A&M University-Kingsville (#05272021/1469), and we have followed the code of practice adopted for the reported experimentation and methodology.

## Supporting information

**S1 Photo. Side-by-side color comparison of Plateau (*Holbrookia lacerata*) and Tamaulipan (*H. subcaudalis*) spot-tailed earless lizards (STEL).** Plateau STEL (right) are caramel colored and Tamaulipan STEL (left) are a slate gray color.
(PNG)

**Supplementary material 1.** Code for statistical analysis of hypothesis 1: testing variation detection probability of lizards across known densities using two different capture methods: road cruising and visual searches.
(DOCX)

## Acknowledgments

We thank the Texas Comptroller of Public Accounts for financial assistance and Texas A&M University-Kingsville for access to property. We thank L. Willard, C. A. Moeller, and B. Moeller for assistance with construction of the enclosure and T. Hanzak of Coastal Bend Canine College for detector dog training. This is contribution number 24-126 of the Caesar Kleberg Wildlife Research Institute.

## Author contributions

**Conceptualization:** Scott E. Henke, Cord B. Eversole.

**Data curation:** Evan Drake Rangel, Scott E. Henke, Cord B. Eversole.

**Formal analysis:** David B. Wester, Gabriel Andrade-Ponce.

**Funding acquisition:** Scott E. Henke, Cord B. Eversole.

**Investigation:** Evan Drake Rangel, Scott E. Henke, Ruby A. Ayala, Cord B. Eversole.

**Methodology:** Evan Drake Rangel, Scott E. Henke, David B. Wester, Ruby A. Ayala, Cord B. Eversole.

**Project administration:** Scott E. Henke, Cord B. Eversole.

**Resources:** Cord B. Eversole.

**Software:** David B. Wester.

**Supervision:** Scott E. Henke, Cord B. Eversole.

**Validation:** David B. Wester.

**Visualization:** David B. Wester.

**Writing – original draft:** Scott E. Henke, David B. Wester, Cord B. Eversole.

**Writing – review & editing:** Scott E. Henke, David B. Wester, Gabriel Andrade-Ponce, Cord B. Eversole.

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
