## [Decision Letter · Decision Letter 0]

1 Jul 2025

Dear Dr. Henke,

Thank you for submitting your manuscript to PLOS ONE. After careful consideration, we feel that it has merit but does not fully meet PLOS ONE’s publication criteria as it currently stands. Therefore, we invite you to submit a revised version of the manuscript that addresses the points raised during the review process.

We look forward to receiving your revised manuscript.

Kind regards,

Edvard Mizsei

Academic Editor

PLOS ONE

Journal Requirements:

Additional Editor Comments (if provided):

I note several areas where the manuscript would benefit from clarification and greater consistency:

Method nomenclature: Please harmonise the names of your survey techniques throughout. For example, lines 220–223 refer to “rock-mound trapping,” whereas elsewhere you use “rock mounds.”

Survey chronology: The temporality of your surveys is difficult to follow. I recommend adding a schematic (e.g. a figure or table) with time on the x-axis and survey method on the y-axis to illustrate the sequence and any overlap of techniques.

eDNA interpretation: Clarify whether and how environmental DNA may have accumulated over the course of the study.

Modelling approaches: I was surprised that the authors did not consider occupancy and n-mixture models, both of which have variants for single- and multiple-visit sampling scenarios (see MacKenzie, Royle & Sólymos). Please discuss why these frameworks were omitted or, preferably, reanalyse your data using one or both methods.

Detectability fundamentals: I suggest incorporating more of the foundational sampling and analytical methods suitable for situations in which detectability is less than one and not constant (see Murray & Sandercock’s Population Ecology in Practice, Fig. 3.1).

Please address these editorial points alongside all comments raised by the reviewers in your revised submission.

Reviewers' comments:

Reviewer's Responses to Questions

**Comments to the Author**

1. Is the manuscript technically sound, and do the data support the conclusions?

Reviewer #1: Yes

Reviewer #2: Yes

2. Has the statistical analysis been performed appropriately and rigorously?

Reviewer #1: Yes

Reviewer #2: Yes

3. Have the authors made all data underlying the findings in their manuscript fully available?

Reviewer #1: No

Reviewer #2: Yes

4. Is the manuscript presented in an intelligible fashion and written in standard English?

Reviewer #1: Yes

Reviewer #2: Yes

Reviewer #1: I am aware of the limitations of the model, species, and space, but some of the biology of the species might be missing from the discussion. STELs normal behaviours might be inhibited due to the translocation, Cooper (2003) found that lizards from that genus change behaviour due to severe changes in body condition (autotomy), they might do it too when translocated, and be less active when that happens, so detectability might be less. Cooper and Guilette (1991) found sexual dimorphic changes in their behaviour too. What sex where the individuals you translocated?

Telemetry, particularly some pit tags, or other short-distance telemetry would be able to answer those kind of questions and let you know if the lizards where there but you couldn't detect them.

Comparing your results to trying to count them on the field, where they naturally live would be a great next step, maybe there the other methods might be more useful, as individuals would be in their natural habitat.

Is it possible that the time needed for eDNA (and chemicals detected by dogs) to be available in sufficient amounts was too short in your study?

The individuals translocated where left there? If so, doesn´t it places an ethical dilemma? Both species do not occur naturally in the same place. Could they reproduce between both of them? Couldn't the project take the 39 living ones back to their natural habitat? It might be important as at least the Tamaulipecan one is endangered.

Aren't numbers of STELs found by both direct systematic search and cross road search too small for statistics to have enough power to bring solid conclusions? I would need to see the model structure, results, residuals and the data, at least a summary for me to be able to reach a conclusion.

I can asume those earless lizards are very sensitive to vibrations in the ground, so could ATVs be bothering or scaring them so they hide resulting in a lower detectability and observation rate, compared to a human walking over a dirt road?

The habitat you translocated them to was different from the one they lived in, so you had to take out the native vegetation and allow something somewhat alike their habitat to grow? Could that be accounted for to explain the low detection ratio?

More attractive and informative figures, including maps would be appreciated, together with STELs pictures, as this is an online only journal they wouldn't cost more, but could enrich the manuscript

I would be delighted to read a revised version of this manuscript

Reviewer #2: The STEL were placed in a novel environment, which may represent the most vulnerable aspect of the experimental design. Any research involving these two species—or closely related ones—regarding animal personality in unfamiliar settings or other ethological factors should be taken into account. Such studies are essential for contextualizing and refining the interpretation of these results. Accordingly, this limitation should be acknowledged in the discussion and explicitly mentioned as a cautionary note in the abstract.

**Do you want your identity to be public for this peer review?** For information about this choice, including consent withdrawal, please see our Privacy Policy

Reviewer #1: **Yes: ** EDGARD DAVID MASON-ROMO, PhD

Reviewer #2: No

---

## [Author Response · Author response to Decision Letter 1]

15 Sep 2025

Comments of Editor and Reviewers and Responses of Authors to Comments

Below are the comments of the Editor and two reviewers to our manuscript. The author(s) responses are listed below each comment and are highlighted in Red.

Editor Comments:

I note several areas where the manuscript would benefit from clarification and greater consistency:

Method nomenclature: Please harmonise the names of your survey techniques throughout. For example, lines 220–223 refer to “rock-mound trapping,” whereas elsewhere you use “rock mounds.”

The manuscript was checked and revised. Method names have been changed, as needed, to maintain consistency throughout the manuscript.

Survey chronology: The temporality of your surveys is difficult to follow. I recommend adding a schematic (e.g. a figure or table) with time on the x-axis and survey method on the y-axis to illustrate the sequence and any overlap of techniques.

A figure has been added to provide the chronology of events for each method at each STEL density. Hopefully the figure provides clarification as to what was done during each survey period.

eDNA interpretation: Clarify whether and how environmental DNA may have accumulated over the course of the study.

Greater explanation of the eDNA results have been provided within the Discussion section and we hope provides clarity as to why eDNA did not appear to accumulate over time as expected.

Modelling approaches: I was surprised that the authors did not consider occupancy and n-mixture models, both of which have variants for single- and multiple-visit sampling scenarios (see MacKenzie, Royle & Sólymos). Please discuss why these frameworks were omitted or, preferably, reanalyse your data using one or both methods.

The goal was to test differences in detectability efficiency among methods, not to estimate species density, which is already known. This is an important point the reviewer may be overlooking. Both occupancy and N-mixture models are designed to estimate abundance or presence while accounting for imperfect detection. Although these models can be used to make inferences about the detection process, this is not appropriate when the sole objective is to model detection itself.

To explain this, it’s useful to briefly describe how hierarchical models are structured. In general, both occupancy and N-mixture models consist of two linked GLMs: one that models the ecological process (abundance or occupancy), and another that models the observational process (detection). In this case, we're specifically interested in the observational component, so there’s no point in adding the ecological component to the equation — since abundance is already known.

Additionally, we wouldn't recommend hierarchical models in this case, as the sample size for each method is too small. In theory, sample sizes below 25, with six repetitions and low detection probability (as seen in the new analysis), can result in high bias in N-mixture models. Each methodology in our study has only five spatial replicates. Moreover, since we experienced convergence issues even with a GLM, hierarchical models would likely perform worse. This concern is supported by previous studies showing that small sample sizes and low detectability can produce unreliable estimates and strong biases in N-mixture models (Ficetola et al. 2018; Duarte et al. 2018).

• Ficetola, G.F., Barzaghi, B., Melotto, A. et al. (2018). N-mixture models reliably estimate the abundance of small vertebrates. Scientific Reports, 8, 10357.

• Duarte, A., Adams, M.J., & Peterson, J.T. (2018). Fitting N-mixture models to count data with unmodeled heterogeneity: Bias, diagnostics, and alternative approaches. Ecological Modelling, 374, 51–59.

Detectability fundamentals: I suggest incorporating more of the foundational sampling and analytical methods suitable for situations in which detectability is less than one and not constant (see Murray & Sandercock’s Population Ecology in Practice, Fig. 3.1).

To address the modeling and detectability concerns, we have added a binomial GLM, so that the response variable reflects detection probabilities. The predictors remain the same — abundance treatments and methodologies — but such a model would strengthen our conclusion because initially we failed to detect differences between road cruising and visual searches, but as pointed out, the power of test was low due to sample size. Adding the binomial GLM has greater power of the test, and with both methods displaying similar conclusions, strengthens the manuscript.

Please address these editorial points alongside all comments raised by the reviewers in your revised submission.

Reviewer's Responses to Questions

1. Is the manuscript technically sound, and do the data support the conclusions?

Reviewer #1: Yes

Reviewer #2: Yes

2. Has the statistical analysis been performed appropriately and rigorously?

Reviewer #1: Yes

Reviewer #2: Yes

3. Have the authors made all data underlying the findings in their manuscript fully available?

Reviewer #1: No Data is available on FigShare – STEL Data file. Some data (i.e., location data) is restricted due to Texas State private properties regulations. New analyses will be added to this file so analyses are publicly available.

Reviewer #2: Yes

4. Is the manuscript presented in an intelligible fashion and written in standard English?

Reviewer #1: Yes

Reviewer #2: Yes

Review Comments to the Author

Reviewer #1: I am aware of the limitations of the model, species, and space, but some of the biology of the species might be missing from the discussion. STELs normal behaviours might be inhibited due to the translocation, Cooper (2003) found that lizards from that genus change behaviour due to severe changes in body condition (autotomy), they might do it too when translocated, and be less active when that happens, so detectability might be less. Cooper and Guilette (1991) found sexual dimorphic changes in their behaviour too. What sex where the individuals you translocated?

We now acknowledge that STEL behavior could have been altered and we have added the Cooper (2003) citation as suggested by the reviewer. We also added a statement of observed behavior during the study that suggests that STEL behavior may not have altered. Also, for greater clarity, we have included a new table that addresses the sex ratios at each density for each species.

Telemetry, particularly some pit tags, or other short-distance telemetry would be able to answer those kind of questions and let you know if the lizards where there but you couldn't detect them.

We placed the STEL into the enclosure at known densities. We also addressed the timeframe of surveys so reproduction was not an issue, and the precautions that were used to reduce the likelihood of predation during the study. Therefore, densities of STEL were known.

Comparing your results to trying to count them on the field, where they naturally live would be a great next step, maybe there the other methods might be more useful, as individuals would be in their natural habitat.

We added statements concerning the habitat within our enclosure and STEL ‘natural’ habitat to clarify that the two are quite similar, so STEL placed within the enclosure was not completely foreign habitat.

In addition, we did road cruising and walking transects for Tamaulipan STEL as the next step, which has been published, and we cite that work within this manuscript.

Is it possible that the time needed for eDNA (and chemicals detected by dogs) to be available in sufficient amounts was too short in your study?

We added statements concerning the eDNA and detector dogs that address climatic problems with longevity.

The individuals translocated where left there? If so, doesn´t it places an ethical dilemma? Both species do not occur naturally in the same place. Could they reproduce between both of them? Couldn't the project take the 39 living ones back to their natural habitat? It might be important as at least the Tamaulipecan one is endangered.

We added the reasoning for maintaining the STEL within the enclosure to the manuscript. In a nutshell, translocation was another component of the overall study. STEL were maintained to determine if translocation could be successful, do translocated STEL reproduce, survival after translocation, etc. (i.e., largest known population of Tamaulipan STEL occur on property that was recently purchased by Elon Musk for a lithium factory to make batteries for his Tesla cars. My fear is that the largest population of Tamaulipan STEL will not last if left on their own.)

Aren't numbers of STELs found by both direct systematic search and cross road search too small for statistics to have enough power to bring solid conclusions? I would need to see the model structure, results, residuals and the data, at least a summary for me to be able to reach a conclusion.

We have added a binomial GLM that has addressed the issue of small sample size. The analyses will be placed in a public repository, and if preferred, can be added as a supplement to this manuscript.

I can assume those earless lizards are very sensitive to vibrations in the ground, so could ATVs be bothering or scaring them so they hide resulting in a lower detectability and observation rate, compared to a human walking over a dirt road?

It is unknown if STEL react, positively or negatively, to ground vibration in terms of detectability. However, road cruising resulted in some of the best detectability so it does not appear that potential ground vibration caused lower detectability than other methods. We included a citation that compares road cruising vs. walking transects for Tamaulipan STEL to highlight this.

The habitat you translocated them to was different from the one they lived in, so you had to take out the native vegetation and allow something somewhat alike their habitat to grow? Could that be accounted for to explain the low detection ratio?

We added statements concerning the habitat within our enclosure and STEL ‘natural’ habitat to clarify that the two are quite similar, so STEL placed within the enclosure was not completely foreign habitat.

More attractive and informative figures, including maps would be appreciated, together with STELs pictures, as this is an online only journal they wouldn't cost more, but could enrich the manuscript

We’ve added additional tables and figures and photos of STEL as requested. Hopefully you find them attractive.

I would be delighted to read a revised version of this manuscript

Reviewer #2: The STEL were placed in a novel environment, which may represent the most vulnerable aspect of the experimental design. Any research involving these two species—or closely related ones—regarding animal personality in unfamiliar settings or other ethological factors should be taken into account. Such studies are essential for contextualizing and refining the interpretation of these results. Accordingly, this limitation should be acknowledged in the discussion and explicitly mentioned as a cautionary note in the abstract.

We have acknowledged that STEL were translocated and how translocation may have altered their normal behavior, so caution should be exercised with interpretation.

---

## [Decision Letter · Decision Letter 1]

21 Oct 2025

Efficacy of various survey methods to detect an experimental population of spot-tailed earless lizards: A case study

PONE-D-25-16961R1

Dear Dr. Henke,

We’re pleased to inform you that your manuscript has been judged scientifically suitable for publication and will be formally accepted for publication once it meets all outstanding technical requirements.

Kind regards,

Edvard Mizsei

Academic Editor

PLOS ONE

Additional Editor Comments (optional):

Reviewers' comments:

Reviewer's Responses to Questions

**Comments to the Author**

Reviewer #1: All comments have been addressed

Reviewer #2: All comments have been addressed

2. Is the manuscript technically sound, and do the data support the conclusions?

Reviewer #1: Yes

Reviewer #2: Yes

3. Has the statistical analysis been performed appropriately and rigorously?

Reviewer #1: Yes

Reviewer #2: Yes

4. Have the authors made all data underlying the findings in their manuscript fully available?

Reviewer #1: Yes

Reviewer #2: Yes

5. Is the manuscript presented in an intelligible fashion and written in standard English?

Reviewer #1: Yes

Reviewer #2: Yes

Reviewer #1: All comments have been properly answered, thanks for your patience and politeness. I think this is a valuable contribution

Reviewer #2: (No Response)

**Do you want your identity to be public for this peer review?** For information about this choice, including consent withdrawal, please see our Privacy Policy

Reviewer #1: **Yes: ** Edgard David Mason Romo

Reviewer #2: No

---

## [Editor Report · Acceptance letter]

PONE-D-25-16961R1

PLOS ONE

Dear Dr. Henke,

I'm pleased to inform you that your manuscript has been deemed suitable for publication in PLOS ONE. Congratulations! Your manuscript is now being handed over to our production team.

Kind regards,

on behalf of

Dr. Edvard Mizsei

Academic Editor

PLOS ONE